# Prognostic and Predictive Effects of Tumor and Plasma miR-200c-3p in Locally Advanced and Metastatic Breast Cancer

**DOI:** 10.3390/cancers14102390

**Published:** 2022-05-12

**Authors:** Esther Navarro-Manzano, Ginés Luengo-Gil, Rocío González-Conejero, Elisa García-Garre, Elena García-Martínez, Esmeralda García-Torralba, Asunción Chaves-Benito, Vicente Vicente, Francisco Ayala de la Peña

**Affiliations:** 1Department of Hematology and Medical Oncology, Hospital Universitario Morales Meseguer, 30008 Murcia, Spain; esther.navarro3@um.es (E.N.-M.); ginesluengo@hotmail.com (G.L.-G.); rocio.gonzalez@carm.es (R.G.-C.); eliggarre3@yahoo.es (E.G.-G.); helenagarciam@gmail.com (E.G.-M.); esme_bk@hotmail.com (E.G.-T.); vicente.vicente@carm.es (V.V.); 2Centro Regional de Hemodonación, 30003 Murcia, Spain; 3Instituto Murciano de Investigación Biosanitaria, IMIB, 30120 Murcia, Spain; 4Department of Medicine, Medical School, University of Murcia, 30001 Murcia, Spain; mariaa.chaves@carm.es; 5Medical School, Universidad Católica San Antonio, 30107 Murcia, Spain; 6Department of Pathology, Hospital Universitario Morales Meseguer, 30008 Murcia, Spain

**Keywords:** breast cancer, miR-200c, neoadjuvant chemotherapy, plasma biomarkers, prognostic factor, predictive factor

## Abstract

**Simple Summary:**

The miR200 family is involved in breast cancer progression. Our aim was to evaluate the predictive and prognostic role of miR-200c-3p, both in plasma and in tumor, in women with locally advanced and metastatic breast cancer. Our results show that plasma levels of miR-200c-3p are higher in these patients and might be used as a breast cancer marker. In patients treated with neoadjuvant chemotherapy, miR-200c-3p expression may also improve prognostic stratification beyond pathologic response and clinical stage.

**Abstract:**

While the role of miR-200c in cancer progression has been established, its expression and prognostic role in breast cancer is not completely understood. The predictive role of miR-200c in response to chemotherapy has also been suggested by some studies, but only limited clinical evidence is available. The purpose of this study was to investigate miR-200c-3p in the plasma and primary tumor of BC patients. The study design included two cohorts involving women with locally advanced (LABC) and metastatic breast cancer. Tumor and plasma samples were obtained before and after treatment. We found that miR-200c-3p was significantly higher in the plasma of BC patients compared with the controls. No correlation of age with plasma miR-200c-3p was found for controls or for BC patients. MiR-200c-3p tumor expression was also associated with poor overall survival in LABC patients treated with neoadjuvant chemotherapy, independently of pathological complete response or clinical stage. Our findings suggest that plasmatic miR-200c-3p levels could be useful for BC staging, while the tumor expression of miR-200c-3p might provide further prognostic information beyond residual disease in BC treated with neoadjuvant chemotherapy.

## 1. Introduction

Breast cancer (BC) is the most prevalent neoplasia among women in developed countries. Despite recent improvements in its diagnosis and treatment, it is still among the leading cancer-related causes of death. Since most cases are diagnosed at early stages, the identification of prognostic factors for metastatic relapse is key for stratifying patients and individualizing therapeutic strategies [1,2]. Besides the acknowledgment of BC intrinsic subtypes as the basis for its heterogeneous biology and clinical behavior, many studies have focused on understanding the biological basis of tumor progression and metastasis and on the identification of biomarkers associated with these processes [3].

MicroRNAs (miRNAs) are involved in multiple human pathological processes, including development, differentiation, cellular proliferation, programmed cell death, cancer initiation and metastasis. The deregulation of the expression of these small molecules in tumor tissue and liquid biopsy has led cancer research towards the search for miRNAs with prognostic and diagnostic utility. MiRNAs are small non-coding RNAs that can downregulate the gene expression of target sequences through mRNA degradation and/or translation blockage. In recent years, some research groups have demonstrated that both normal and cancer cells are able to synthesize and secrete miRNAs to extracellular media via several mechanisms [4,5]. In contrast to the high sensitivity of mRNAs to degradation and structural instability, miRNAs are resistant to nuclease degradation, have a long plasmatic half-life [6,7] and they survive aggressive handling conditions such as tissue fixation and paraffin embedding, conditions in which other molecules would degrade [8]. Moreover, several miRNAs have tissue- or even cell-specific expression, making them ideal candidates for screening as potential cancer biomarkers and even for monitoring disease relapse or progression [9,10,11].

The miR-200 family is among the most widely studied miRNAs in cancer. The family consists of the following five conserved members: miR-200a, miR-200b, miR-200c, miR-141 and miR-429. These miRNAs can participate in epithelial–mesenchymal transition, cell senescence, survival, and other key processes. Specifically, miR-200c has been associated with BC development and progression in many studies. Current evidence shows that miR-200c and other members of the miR-200 family downregulate the expression of ZEB1 and ZEB2, which are transcriptional repressors of CDH1 and are involved in the epithelial to mesenchymal transition processes. The modification of the tumor microenvironment through the direct targeting of Sec-23a and its derived changes on secretome is also a relevant effect of miR-200 [12]. However, the conclusions reached in the studies focused on the prognostic role of miR-200c expression in both tissues and/or plasma in breast cancer are contradictory. Some reports have suggested oncosupressive roles [13,14,15], while other studies indicate that a higher expression of miR-200c might induce, rather than prevent, metastases formation [16]. Xu et al. reported a decreased expression of the miR-200 family in BC associated with lymph-node metastasis [17]. These results are also coherent with a study conducted by Song et al., which correlated the low expression of miR-200c with poor patient overall survival (OS) and disease-free survival (DFS) [18]. While these studies support a tumor-suppressor role for miR-200c, other groups reported high levels or miR-200c in BC tissues and plasma compared with controls. Some of these studies demonstrated the association of miR-200 up-regulation with BC progression and death. Tsai et al. found that miR-200a,c and miR-141 are up-regulated in BC tissue independently of tumor subtype or age group [19]. Fontana et al. evaluated miR-200c expression in tumor tissue both in their own cohort and in a second cohort from The Cancer Genome Atlas (TCGA) and they found that overall, the miR-200c family was up-regulated in breast tumors compared with the non-malignant areas of margins [20]. Furthermore, other studies—including those published by Papadaki et al. and Madhavan et al.—confirmed higher levels of plasmatic miR-200c in metastatic breast cancer (MBC) compared to locally advanced breast cancer (LABC) and controls [21,22]. Additionally, Madhavan et al. documented the ability of miR-200c and other micro-RNAs to serve as surrogate markers of circulating tumor cells (CTCs) in MBC patients. Higher levels of miR-200c and other members of the miR-200 family were associated with worse survival in MBC patients and outperformed CTCs as prognostic markers [23]. These results are also supported by a recent metanalysis of multiple tumors showing data consistent with the association of higher levels of miR-200 and worse prognosis [24].

A new promising area under investigation is the use of miRNAs as predictive biomarkers of chemoresistance or even as therapeutic targets [25,26]. Moreover, mounting evidence indicates the existence of similarities between drug-resistant and metastatic cancer cells in terms of resistance to apoptosis and enhanced invasiveness. In this framework, recent studies have suggested that miR-200c, among others, would influence the response to chemotherapy in several tumor types, including BC [27,28,29]. Again, data on the role of miR-200c in a patient’s resistance to chemotherapy and endocrine-based therapy are not clear, with preclinical data mostly showing an association of miR-200 down-regulation with chemoresistance [27,30,31,32,33] and endocrine resistance [34,35]. Although clinical data are, as previously stated, available for the prognostic impact of miR-200c in breast cancer, the predictive value of the miR-200 family is less clear and only limited clinical data exist for its correlation with tumor responses [36].

The controversial results for BC and the lack of clinical data on tumor response led us to perform an analysis of miR-200c-3p expression both in plasma samples and in primary tumor tissue (in both samples pre- and post-chemotherapy) in LABC and MBC patients. Our aim was to clarify the role of this microRNA in BC progression, its specific association with clinical features and its response to antineoplastic treatment.

## 2. Materials and Methods

### 2.1. Study Design

In this study, we evaluated miR-200c-3p expression in two cohorts including women with locally advanced breast cancer (LABC) and metastatic breast cancer (MBC). The two cohorts only differed in terms of the time of sample collection. Thus, cohort A included pre- and post-neoadjuvant chemotherapy plasma and biopsies samples from women with LABC. In turn, cohort B was composed of plasma samples from both LABC and MBC patients. The experimental flowchart of our study is outlined in Appendix A. Since patients from cohorts A and B were included in different time periods, some analyses were performed independently for each cohort and for each disease setting.

### 2.2. Clinical Diagnosis and Treatment of Breast Cancer Patients

All LABC patients (cohorts A and B) were diagnosed and treated according to usual clinical practice, following international guidelines. In locally advanced tumors (cT3N1, cN2-3 or cT4), breast MRI and ultrasound examination were included both in the pre- and post-chemotherapy evaluation. Body computed tomography and bone scintigraphy were also added to the staging workup. Preoperative chemotherapy included both taxanes and anthracyclines [37,38]. TheNSABP-B27 regimen was used most frequently and included cyclophosphamide (600 mg/m2/21 days) and doxorubicin (60 mg/m2/21 days) for four courses, followed by docetaxel (100 mg/m2/21 days) for four cycles. Trastuzumab was administered both in the neoadjuvant and the adjuvant setting to those patients whose tumors overexpressed HER2. Adjuvant radiation therapy and endocrine therapy were administered according to current guidelines.

In MBC patients, staging included body computed tomography and bone scintigraphy, complemented when needed with MRI and 18F-FDG PET-TC. First-line treatment was decided according to current clinical practice guidelines and included endocrine-based therapy (mostly aromatase inhibitors) and chemotherapy (either taxanes or capecitabine) with or without biological agents such as trastuzumab or bevacizumab. Clinical response was evaluated according to the RECIST 1.1 criteria [39].

### 2.3. Plasma Levels of miR-200c-3p 

Blood was drawn and collected before and after NCT from patients with LABC (75 and 25 plasma samples, respectively) and before and after the first line of endocrine-based therapy or chemotherapy from MBC patients (42 and 19 plasma samples, respectively). For LABC patients, the time of pre-treatment plasma collection was immediately before starting neoadjuvant chemotherapy (usually the same day); for the post-treatment sample, it was obtained after completing the full course of NCT, three to four weeks after the last cycle and just before surgical treatment. For MBC patients, pre-treatment plasma samples were obtained just before starting the first line of treatment, usually the same day of chemotherapy or a few days before oral endocrine-based treatment was initiated. Post-treatment samples in MBC patients were obtained at the time of the first response evaluation, usually between 8 and 12 weeks after initiating the treatment. We also collected blood from 28 healthy women as controls at the same time that cohort B was recruited. Plasma was prepared by centrifugation for 15 min at 2500 rpm and stored at −80 °C until used. Samples were stored with the Biobank Biobancmur-Node3 (National Biobank registry reference B.0000859), integrated in the Spanish National Biobanks Network–ISCIII.

### 2.4. Pathology

Tissue samples were collected before and after NCT in LABC patients. Pre-treatment estrogen (ER) and progesterone receptors (PR) status was assessed with immunohistochemistry (IHC), and HER2 status was assessed by either fluorescent in situ hybridization or with a validated IHC method (Hercep Test, Dako North America Inc., Carpinteria, CA, USA). For ER and PR, cases were considered negative when the percentage of immunoreactive tumor cells was below 1%; the rest of the cases (≥1% of tumor cells stained) were classified as positive. For HER2, cases were considered positive if the Hercep Test result was 3+ and/or if FISH showed a ratio HER2/CEP17 ≥ 2; the rest of the cases were classified as negative. Tumors were phenotypically classified according to pre-treatment IHC results as hormone receptors (HR) that were positive HER2 negative (ER and/or PR positive and HER2 negative), HR positive HER2 positive (ER and/or PR positive and HER2 positive), HR negative HER2 positive (ER and PR negative and HER2 positive) or triple negative (ER negative and PR negative and HER2 negative). TNM staging was determined according to the 8th edition AJCC system. In LABC cases treated with NCT, a pathological complete response (pCR) was defined as the absence of an invasive carcinoma both in the breast and the axilla, regardless of the presence of carcinoma in situ (ypT0/Tis ypN0). 

### 2.5. miRNA and mRNA Isolation and RT-qPCR Analysis

Small RNA from plasma samples were extracted using NucleoSpin miRNA Plasma (Macherey-Nagel, Dueren, Germany) according to the manufacturer’s instructions. Plasma samples presenting hemolysis were not processed for further analysis. Breast cancer tissues from FFPE samples (pre- and post-chemotherapy), which are considered suitable samples for miRNA analysis [40] were used for the study. Cylindrical punches (1.5 mm) were obtained, and after the identification of tumor areas (>50% tumor cellularity) by a pathologist, samples were deparaffinized using xylene and ethanol washes and then RNA was isolated using the miRNeasy FFPE Kit in a QIAcube robotic workstation system (QIAgen) following the supplier’s protocol. RNA retrotranscription and qPCR were performed using Takara Premix Taq (Clontech/Takara Bio, San Jose, CA, USA) and miRNA Taqman assays (Life Technologies, Darmstadt, Germany) in a LightCycler 480 System (Roche). Each qPCR plate included technical replicates, positive and negative controls of reverse transcription and multiple water blanks. Relative expression was calculated using snU6 and miR-16 as the endogenous controls for tissue and plasma samples, respectively, with the 2-DeltaDelta Ct normalization method. RNA levels in post-treatment tissue were analyzed only in patients without primary tumor pCR (with residual tumor cells present in FFPE samples).

The total RNA from tumor samples was extracted using an RNeasy FFPE Kit (QIAGEN) following the supplier’s instructions. All mRNAs (with preamplification) were retrotranscribed and amplified using TaqMan^®^ Gene Expression Assays (Life Technologies, Darmstadt, Germany) in the same LightCycler^®^ 480 Real-Time PCR System. Relative expression was calculated with the 2ΔCt method, using ACTB as the endogenous control mRNA.

### 2.6. Statistical Methods

Patient’s clinicopathological characteristics were reported as median along the interquartile range (IQR) or as frequencies and percentages for continuous and categorical variables, respectively. For miR-200c-3p plasma-expression values, probable outliers were defined according to Tukey’s method as values below or over (Q1-3 − 1.5 IQR, Q1-3 + 1.5 IQR) and were excluded from the analysis. The normal distribution assumption of miRNA expression was evaluated with the Kolmogorov–Smirnov and Shapiro–Wilks tests. MiR-200c-3p levels were dichotomized using the median value as the cut-off. The association of miR-200c-3p expression with clinical and pathological characteristics and the analysis of paired samples were evaluated with non-parametric tests. The correlation of miR-200c-3p expression and the expression of other miRNAs or miRNA-related genes was evaluated with Spearman’s rank correlation test. Kaplan–Meier curves, log-rank test and Cox univariable and multivariable proportional hazard regression models were used for disease-free survival and overall survival analyses. MiR-200c-3p median pre-NCT and post-NCT expression was arbitrarily chosen as a predefined cut-off point for survival comparisons. The statistical analysis was performed with IBM SPSS 21.0 software and with R version 4.1.3.

## 3. Results

### 3.1. Patients and Tumor Characteristics

One hundred and seventy-one patients with LABC treated with neoadjuvant chemotherapy (NCT) were included in the study. Clinical and pathological features are shown in Table 1 and data for each of the two LABC cohorts (A and B) are provided in Appendix A. Except for the age and frequency of the HER2+ HR+ subtype, there were no significant differences in the clinical characteristics between the two cohorts. For all LABC patients, the median age was 56 years (range from 21 to 79). Stages were mostly IIB (28.5%) or IIIA (33.3%). The predominant histology was invasive ductal carcinoma (95.9%) and histological grade 3 (49.1%). Most patients (45.6%) had hormone receptor (HR)-positive/Human epidermal growth factor receptor 2 (HER2)-negative tumors, and 22.8% had triple-negative breast cancer (TNBC). The pathological complete response (pCR) rate was 18.9%. Tumor phenotype together with histological grade predicted pCR in a logistic-regression multivariate model (Appendix A). With a median follow up of 132 months, 38 patients showed recurrence (33 distant relapses, 5 isolated local relapses) and 31 deaths occurred. Neither overall survival (OS) nor disease-free survival (DFS) were reached.

A control group of 28 women without relevant comorbidities and with no previous neoplasms was also recruited for plasma studies; the median age was 34 (range, 22–59).

Table 2 summarizes the clinicopathological characteristics of the 42 MBC patients included in the study. Median age was 59 years and the performance status was good in most women. Most patients (78.6%) had de novo MBC (stage M1) and the most frequent subtype was HR+/HER2− (73.8%). Bone was the predominant metastatic location, although 45.2% of the patients had visceral disease. With a median follow-up of 80 months, 33 deaths occurred, and median OS was 50.4 months (95% CI, 38.9–61.8). Detailed first-line treatment is shown in Appendix A.

### 3.2. Plasma Levels of miR-200c-3p in Breast Cancer Patients and Controls

We evaluated the expression profile of miR-200c-3p in plasma from LABC (cohort B; *n* = 62) and MBC patients (*n* = 42) as well as in the plasma from the control group (*n* = 28). Plasma expression data for each group and cohort are shown in Appendix A. In samples collected pre-NCT, the levels of miR-200c-3p were significantly higher in LABC and MBC compared with the controls (*p* = 0.034 and *p* = 0.008, respectively), although the differences between the two groups of patients were not statistically significant (*p* = 0.429) (Figure 1). A combined analysis of both cohorts showed similar results (data not shown). Interestingly, in post-treatment plasma samples, the levels of miR-200c-3p were significantly higher in MBC than in LABC patients (Mann–Whitney U test *p* = 0.003) (Appendix A).

An exploratory analysis of the potential utility of miR-200c-3p as a potential diagnostic test for BC (LABC or MBC) showed a receiver operating characteristic curve (ROC) AUC of 0.66 (95% CI, 0.56–0.75).

### 3.3. miR-200c-3p Plasma Levels and Clinicopathological Features in Locally Advanced Breast Cancer

To analyze the correlation between miR-200c-3p plasma levels and tumor dissemination, we first evaluated the association of miR-200c-3p with nodal disease in LABC patients with both tumor and plasma (*n* = 12) samples available (cohort A). As shown in Figure 2a, in plasma samples pre-NCT, the levels of miR-200c-3p were significantly higher in those patients with advanced nodal involvement (Mann–Whitney U test, *p* = 0.016). However, this association was not observed in the post-chemotherapy samples (*p* = 0.812; Figure 2b).

To confirm these results, we also analyzed the association of nodal metastasis and plasma levels of miR-200c-3p in combined LABC cohorts A and B (*n* = 73) after the normalization of miRNA expression values in both cohorts. Although we found a trend towards higher plasmatic levels in women with cN2-3 disease (*n* = 22), this difference was not statistically significant (*p* = 0.059) (Appendix A). Similarly, plasmatic levels of miR-200c-3p were not associated with stage, subtype, grade or other relevant clinic-pathological characteristics of LABC patients (data not shown).

Since the age distribution was significantly different between controls and BC patients (*p* < 0.001), we analyzed the correlation of age with miR-200c-3p plasma levels. A correlation with age was not found for LABC patients (r = 0.076, *p* = 0.52) or for the MBC group (r = −0.084, *p* = 0.605). The same results were obtained for post-treatment levels both in the LABC (r = 0.044, *p* = 0.833) and the MBC group (r = −0.073, *p* = 0.772). No correlation was observed between age and miR-200c levels in the control group (r = −0.034; *p* = 0.867).

### 3.4. Prognostic Role of miR-200c-3p Plasma Levels

Plasma levels of miR-200c-3p showed no clear association with OS in LABC patients (pre-NCT levels, *p* = 0.599; post-NCT levels, *p* = 0.480) (Appendix A). Similarly, no differences were found for DFS according to pre-NCT (*p* = 0.296) or post-NCT (*p* = 0.301) miR-200c-3p plasma levels (Appendix A). The results for the whole group were maintained when the analysis was performed independently in each LABC cohort, with a non-significant impact of either pre-treatment or post-treatment-circulating miR-200c-3p levels on overall survival or disease-free survival (*p*-values NS for all comparisons).

The analysis of the pre- and post-treatment plasma levels of miR-200c-3p in the MBC group did not show any association with OS (pre-treatment, *p* = 0.160; post-treatment; *p* = 0.842) or progression-free survival (pre-treatment, *p* = 0.794; post-treatment, *p* = 0.167). Similarly, no association was found for the pre-treatment (*p* = 0.780) or post-treatment (*p* = 0.626) plasma levels of this miRNA with an objective response to anticancer therapy.

### 3.5. miR-200c-3p Expression in Tumor and Clinicopathological Features in Locally Advanced Breast Cancer Patients

We wanted to evaluate the relationship between miR-200c-3p expression in tumor and the clinical features of patients. For this, 96 paraffin-embedded pre-treatment core biopsies and post-treatment surgical biopsies from the LABC cohort A were available. We first evaluated the expression of miR-200c-3p and other members of the miR-200 family. A highly positive correlation was found for the expression of miR-200c-3p with miR-141 (r = 0.585, *p* < 0.001) and miR-429 (r = 0.635, *p* < 0.001); further analyses exclusively focused on miR-200c-3p. In these samples, pre-NCT miR-200c-3p expression was associated with extensive (cN2-3) clinical nodal involvement (Mann–Whitney U test, *p* = 0.024; Figure 2c). After chemotherapy, this association was maintained, with higher miR-200c-3p expression in the primary tumor for ypN-positive patients (Mann–Whitney U test, *p* < 0.001; Figure 2d).

No clear associations were observed between the miR-200c-3p expression level and grade, stage, or tumor phenotype at diagnosis in the primary tumor (Appendix A). Additionally, no association was found between pre-NCT miR-200c-3p expression and pCR, suggesting that miRN-200c expression is not involved in resistance to chemotherapy.

### 3.6. Prognostic Role of miR-200c-3p Tumor Expression in Locally Advanced Breast Cancer Patients

We first analyzed the impact of primary tumor miR-200c-3p expression on the survival of LABC patients. Median miR-200c-3p tumor expression pre-NCT (0.2677) and post-NCT (0.2607) were arbitrarily predefined as the cut-off points to distinguish the following two groups of patients: miR-200c-3p high and low. High pre-treatment expression of miR200-3p was significantly associated with worse OS (*p* = 0.027) (Figure 3a). This prognostic association was also observed in post-NCT tumor samples, with worse OS in patients with a high expression of miR-200c-3p in the residual tumor (*p* = 0.015) (Figure 3b). The prognostic ability of miRNA levels was confirmed in a multivariate Cox proportional hazard regression model including other clinically relevant covariates, both in the pre-NCT (HR 3.98, 95% CI 1.49–10.60; *p* = 0.006) and in the post-NCT setting (HR 2.63, 95% CI 1.01–6.86; *p* = 0.049) (Table 3).

On the other hand, in the univariate model, DFS was associated with miR-200c-3p tumor expression neither before *p* = 0.134) nor after NCT (*p* = 0.105) (Table 3; Figure 3c,d). Regarding distant relapse-free survival (DRFS) and miR-200c-3p tumor expression, we found a trend towards statistical significance in pre-NCT samples (*p* = 0.074), whereas it reached statistical significance in post-NCT samples (*p* = 0.024) (Figure 3e,f), probably explaining the impact of miRNA expression on OS. In the multivariate analysis, neither pre-NCT nor post-NCT miR-200c-3p tumor expression were statistically significant for DRFS (Table 3).

### 3.7. Correlation between Plasma and Tumor Expression of miR-200c-3p in Locally Advanced Breast Cancer Patients

Finally, we determined the association between levels of miR-200c-3p in primary tumor and in plasma in a small group of LABC (cohort A) patients in which both samples were collected at the same time before NCT (*n* = 12). Our analysis showed that there was no correlation between baseline tissue and plasma miR-200c-3p expression (Rho = 0.217; *p* = 0.49).

### 3.8. Changes in miR-200c-3p-Dependent Proliferation and EMT Gene Expression

To evaluate the potential effect of miR-200c-3p on key processes related to tumor progression and metastases generation, we analyzed gene expression changes associated with miR-200c-3p expression in the tumor of LABC patients before NCT. As shown in Table 4, the expression of MKI67, MYBL2 and CCNB1 was significantly correlated with miR-200c-3p expression, both in pre- and post-NCT tumor samples. EMT-related genes also showed a weak correlation with miR-200c-3p expression, with a significant inverse correlation for VIM and a trend towards a higher expression of CDH1. 

We also analyzed the expression of two genes potentially involved in response to taxanes (TUBB3) and to endocrine-based treatment (ESR1), without finding a relevant correlation of their expression levels with miR-200c-3p expression. A high level of association was found between proliferation-related genes (Figure 4).

The analysis of specific gene expression changes in each BC subtype showed that in luminal (HR+ HER2− BC, the most frequent subtype of BC, the expression of miR-200c-3p was associated with a profile characterized by enhanced proliferation, the significant inhibition of VIM and higher expression of both CDH1 and ESR1. No clear profile was evident in HER2-positive tumors, except for an association with a lower expression of TUBB3. In TNBC patients, the correlation of miR-200c-3p expression with proliferation-related genes was significantly stronger, while no changes were observed for EMT-related or treatment-resistance-related genes.

Finally, we analyzed the post-treatment association of miR-200c-3p expression with the expression of the same genes, excluding those patients with a complete pathologic response. A mild association with proliferation-related genes was maintained while the significant correlation with VIM and CDH1 expression was stronger than in the pre-chemotherapy setting (Table 4). Since the sample size was smaller (*n* = 77), we did not perform a detailed analysis by tumor subtype in the post-treatment samples.

## 4. Discussion

The tumor expression and plasma levels of miR-200 have been associated with BC progression and response to treatment. Here, we performed a clinical study to evaluate the predictive and prognostic role of miR-200c-3p in both primary tumor and plasma. We also provided data on miR-200c-3p tumor expression in the context of anthracyclines and taxanes-based neoadjuvant chemotherapy for BC. The availability of pre- and post-treatment samples of primary tumor from formalin-fixed paraffin-embedded tissue sections allowed us to evaluate both the prognostic and predictive value of miR-200c-3p in the setting of NCT. The plasma levels of miR-200c-3p were also evaluated in healthy controls and in MBC patients, allowing us to identify its potential value as a tumor-related biomarker for breast cancer.

Our data, showing higher plasmatic levels of miR-200c-3p in LABC and MBC patients, are in agreement with previously published data [21,23,41]. Zhang et al. also described a predictive role of increased levels of plasma miR-200c-3p for metastasis in patients without metastatic breast cancer at diagnosis [41]. However, whether elevated levels of miR-200c-3p are a surrogate marker or on the contrary have a role in promoting metastasis is a question that has not yet been answered with certainty. The miR-200 family is considered a marker of epithelial cells, especially if they express E-cadherin [42], suggesting that miR-200 plasma levels might be associated with the circulating epithelial tumor cell load. In fact, the association of higher plasma levels of the miR-200 family with circulating tumor cells (CTCs) in MBC has been previously demonstrated [21,23]. However, beyond this potential explanation as a surrogate for tumor dissemination, the promoting effect on tumor colonization at distant locations of higher levels of miR-200c-3p derived from CTCs has been described in experimental models both for free and exosomal miR-200 [12,43]. Neither Zhang’s work nor our work evaluated the content of miR-200c-3p in patient plasma exosomes, so it is a promising area that should be explored in future research. Ultimately, our results reinforce the association already described between elevated plasma levels of this miRNA and metastatic disease. The fact that different publications have used different miRNA quantification methods and different series of patients reinforces the biomarker role of miR-200c-3p in these patients. Regarding the potential clinical application of these findings for BC diagnosis, our data show an insufficient level of single circulating miR-200c-3p to classify it as a biomarker of BC, although further exploration, perhaps in combination with other circulating biomarkers, might be warranted [23].

Our results showing the association of a higher tumor expression of miR-200c-3p in LABC with a worse prognosis for OS are in agreement with previous publications in other tumors [24], but apparently are in conflict with experimental data showing the role of the miR-200 family as a tumor suppressor in BC. MiR-200 upregulation has been related to the repression of transforming growth factor-beta (TGFb)-induced Epithelial–Mesenchymal Transition (EMT) programs, consistent with non-invasive behavior in ductal in situ carcinomas [44], and with tumor suppressive effects in invasive BC, especially in the setting of TNBC [45,46,47,48]. Additional anti-angiogenic [49] and immune-modulating effects [50] of miR-200c have also been reported in BC. However, other authors have reported opposite effects for the miR-200 family, with miR-200c/141 cluster overexpression causing increased SerpinB2 expression and increased nodal and lung metastases [16]. Our results regarding the association between miR-200c-3p tumor expression and changes in the expression of other genes are, on the whole, consistent with previous works [16,51] showing that miR-200c might be a relevant factor for regulating the balance between proliferation and epithelial–mesenchymal transition (EMT) in epithelial tumors. Although we did not obtain expression data for ZEB1 or other potential direct targets of miR-200c-3p mediating its effects on EMT, both the significant association with a lower expression of vimentin (VIM) and the trend towards a positive association with E-cadherin expression (CDH1) might be consistent with a positive effect on the inverse mesenchymal to epithelial transition, a key step for metastatic colonization. However, more extensive expression data should be obtained to completely determine the effect of miR-200c on EMT transcriptional programs. The up-regulation of proliferation-related genes (MYBL2, KI67, CCNB1) is stronger and supports the pro-proliferative effect as the most likely explanation for the poorer prognosis and the higher potential for metastasis in patients with a higher baseline expression of miR-200c-3p. Taken together, these results support the role of miR-200c as a positive regulator of proliferation and as a suppressor of EMT.

The differential role in different BC subtypes might be another explanation for the previously reported conflicting data, with predominantly oncogenic effects in the luminal subtype through enhanced proliferation, mammosphere formation, lung colonization and PIK3-AKT-signaling activation [52]. Differential expression is not a likely explanation since miR-200c and miR-429 have been reported as being equally expressed across all BC subtypes [20,41]. In fact, we neither found a differential expression of miR-200c-3p related to tumor phenotype, nor observed a different prognostic effect of miR-200c-3p expression (pre- or post-chemotherapy) according to tumor subtype. However, in our work, the analysis of changes in proliferation and EMT-related genes supports the role of miR-200c as a positive regulator of proliferation across all subtypes (with the possible exception of HER2+ disease), while the suppression of EMT seems more relevant in the luminal subtype.

The differential effect of miR-200c in different stages of neoplastic progression might also justify its dual role in metastasis generation; while the miR-200b/a/429 cluster seems to act as a tumor suppressor of metastasis at any stage, the miR-200c/141 cluster might act as a metastasis suppressor at early tumor stages and as a metastasis promoter in the final steps of the metastatic cascade [41]. In our cohort, the prognostic value of tumor miR-200c-3p in tumor in LABC was confirmed not only in the univariate analysis, but also in a multivariate model including pCR and the clinical nodal stage. Although our study cannot completely answer the question of the biological role of miR-200c in BC, taken together, our results suggest that its oncogenic role prevails over its suppressor effects in the final (clinically evident) steps of tumor progression. Our findings might be particularly relevant in the post-chemotherapy setting, in which new biomarkers, beyond pCR, are needed to properly guide new adjuvant strategies [53,54].

A further comment should be made on the correlation between the plasmatic and tumor expression of miR-200c-3p. Our finding of a higher tumor expression of miR-200c-3p in patients with LABC and advanced nodal disease (cN2-3) was paralleled by its plasma levels, which was also higher for this group of LABC patients. However, such differences were not statistically significant. Both the limited sample size and the lack of correlation between plasma and tumor expression in the small group of patients with both available samples, in agreement with previously reported data [41], warrant further research on the integration of tumor and plasma data in a larger series of patients.

The prediction of sensitivity to chemotherapy in breast cancer has been an elusive objective and no clear, single predictive biomarker, beyond the intrinsic subtype and genomic expression test, are available for decision making in the adjuvant or neoadjuvant setting. The predictive role of miR-200c in response to chemotherapy has been suggested by experimental data [30,31] but, to our knowledge, no clinical demonstration of response prediction in the LABC or the MBC settings has been reported. Here, we provided data clearly showing that differential miR-200c-3p expression is not associated with the pathological response in a cohort of LABC patients homogeneously treated with anthracyclines and taxanes-based NCT. The lack of predictive value of plasma miR-200c-3p levels in the first line setting of MBC additionally supports these results, although the heterogeneity of treatments (chemotherapy and endocrine-based therapy) makes this conclusion less concrete in the metastatic setting. Finally, we did not find a clear association between miR-200c-3p expression and the expression of class III beta-tubulin (TUBB3), a potential mediator of resistance to taxane chemotherapy and also an established target of miR-200c. Recent works have pointed to the lower expression of TUBB3 as a potential biomarker of resistance to taxanes [55,56]. The association of higher miR-200c-3p expression with the down-regulation of TUBB3 in HER2-positive disease might warrant further investigation.

Our work has several limitations. First, the analysis of a single miRNA in the miR-200 family might have limited the scope of our findings; although, we found a high correlation between miR-141, miR-429 and miR-200c-3p, and certain data indicate slightly differential effects of the members of this family of miRNAs [12]. Second, although a general profile of enhanced proliferation and repressed EMT seems to be linked to miR-200c-3p overexpression, only some miR-200c-3p-related gene expression changes have been evaluated, limiting the ability to provide further biological explanations for the clinical correlations shown here. Third, the limited sample size for some of the analyses and the combinations of the two independent and slightly heterogeneous cohorts for the plasma results may have limited the statistical power when identifying meaningful clinical differences, especially between BC subtypes. However, we consider that the sample size for the LABC group and the homogeneous treatment of this group both reasonably support the lack of predictive value of miR-200c-3p for the tumor response to NCT. As previously stated, the conclusions are less certain for the MBC setting, in which highly diverse treatments were used, and these results should be considered as exploratory. Another consequence of the limited sample size is the dispersion of miR-200c-3p values for most analyses; however, the aim of this work was to determine the prognostic and predictive value of miR-200c-3p expression, and its clinical implementation would require further work to determine optimal cut-offs and analytical procedures. Finally, age was lower in the control group than in BC patients, which might have biased the comparison of miR-200c-3p plasma levels between them; however, a correlation was found between miR-200c-3p plasma expression and age neither in BC patients nor in the healthy women of the control group. Previous works showing a non-significant trend towards lower miR-200c levels with increasing age [57] also reinforce our results.

## 5. Conclusions

In conclusion, our results support the association of higher plasma levels of miR-200c-3p with both locally advanced and metastatic breast cancer. In the setting of locally advanced breast cancer treated with neoadjuvant chemotherapy (anthracyclines and taxanes), the expression of miR-200c-3p in the tumor microenvironment may also have prognostic relevance for overall survival, independently of tumor subtype, pathological response or clinical stage. However, miR-200c-3p tumor or plasma expression was associated with neither a pathological complete response to neoadjuvant chemotherapy nor with an objective response to first-line treatment in metastatic breast cancer. Taken together, our results negate a predictive role for miR-200c-3p expression, but support its predominantly oncogenic role and its involvement in the metastatic progression of breast cancer. The limited gene expression results are consistent with previous work suggesting that miR-200c-related increased proliferation and repressed EMT are the likely biological basis for these clinical data. The potential application of miR-200c-3p plasma levels as a BC tumor marker for diagnosis and of tumor expression for pre-NCT and post-NCT prognostic stratification should be explored, probably as part of more extensive miRNA profiles.

## Figures and Tables

**Figure 1 cancers-14-02390-f001:**
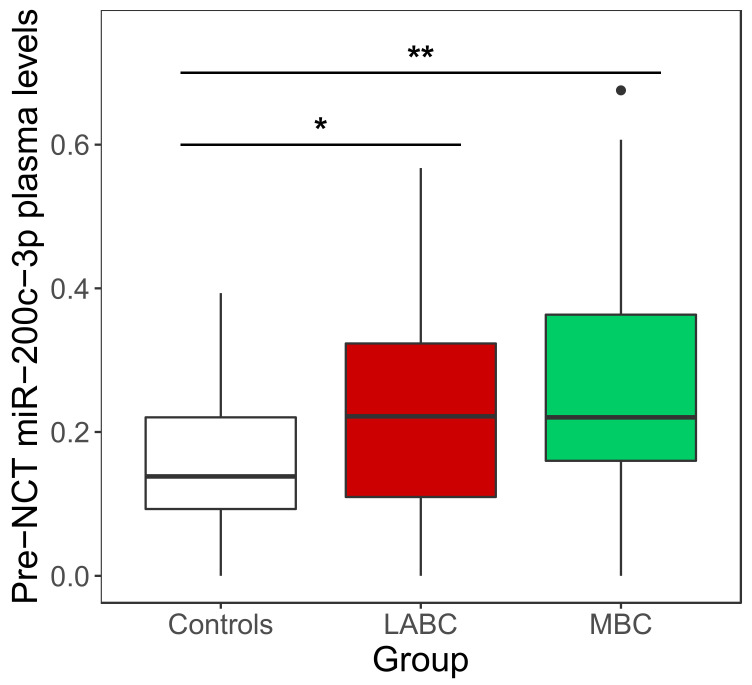
Plasma miR-200c-3p pre-treatment expression in breast cancer patients and controls. Black dots correspond to outliers. Error bars represent ± 1.5 IQR. *, *p* = 0.034; **, *p* = 0.008.

**Figure 2 cancers-14-02390-f002:**
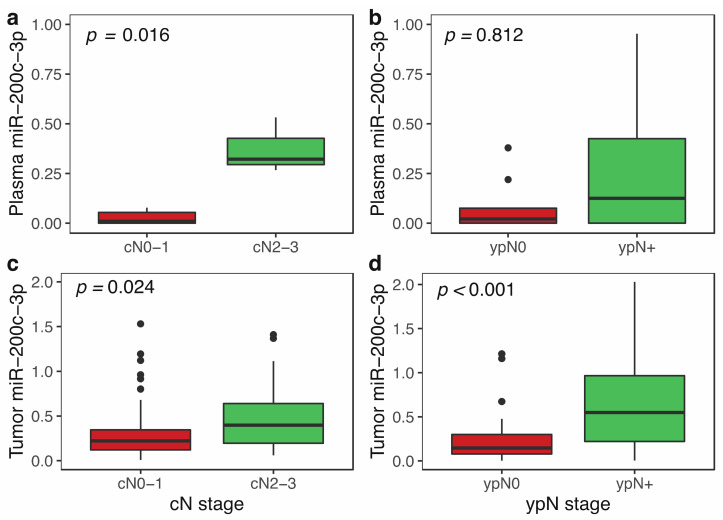
Association of miR-200c-3p expression with clinicopathological features in locally advanced breast cancer (cohort A). (**a**) Pre-NCT plasma levels according to clinical nodal stage (cN0-1 vs. c-N2-3; *p* = 0.016); (**b**) Post-NCT plasma levels according to pathological nodal stage (ypN0 vs. ypN1-3; *p* = 0.812); (**c**) Pre-NCT tumor expression according to clinical nodal stage (cN0-1 vs. c-N2-3; *p* = 0.024); (**d**) Post-NCT tumor expression according to pathological nodal stage (ypN0 vs. ypN1-3; *p* < 0.001). Black dots correspond to outliers.

**Figure 3 cancers-14-02390-f003:**
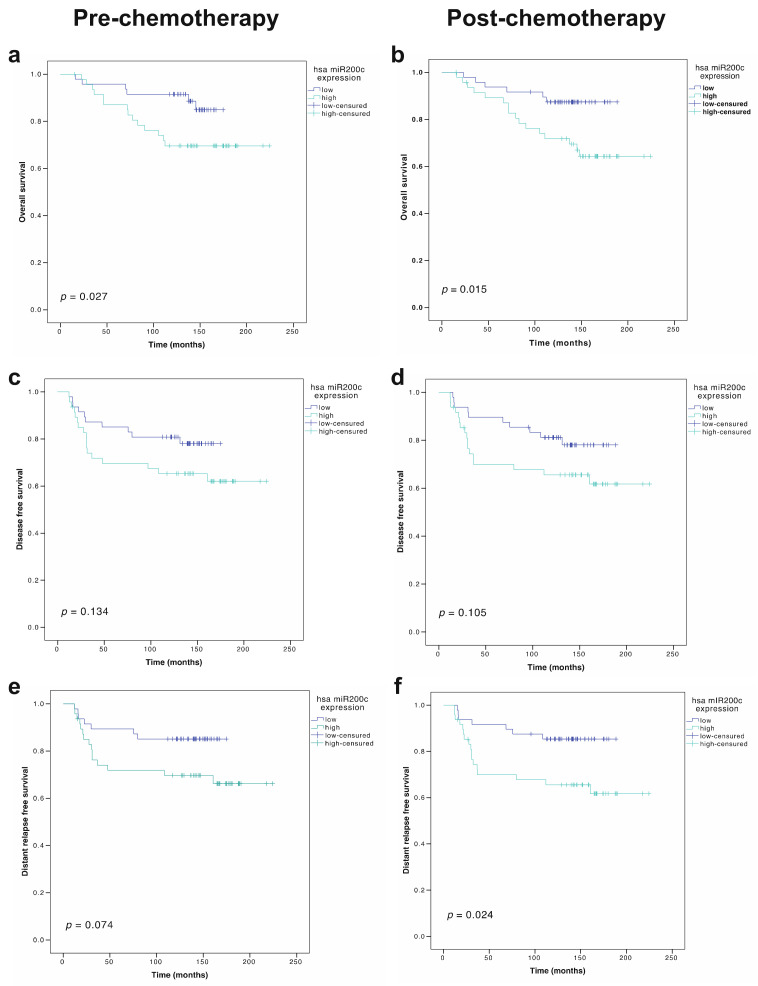
Survival analysis of miR-200c-3p expression in pre- and post-chemotherapy FFPE tumor samples of locally advanced breast cancer patients (cohort A). OS, DFS and DRFS Kaplan–Meier plots according to miR-200c-3p expression in tumor. Median pre-treatment and post-treatment gene expressions were used to define low and high expressers. Univariate Cox regression *p* values are shown in each survival curve. (**a**) OS plot according to pre-chemotherapy miR-200c-3p tumor expression; (**b**) OS plot according to post-chemotherapy miR-200c-3p tumor expression; (**c**) DFS plot according to pre-chemotherapy miR-200c-3p tumor expression; (**d**) DFS plot according to post-chemotherapy miR-200c-3p tumor expression; (**e**) DRFS plot according to pre-chemotherapy miR-200c-3p tumor expression; (**f**) DRFS plot according to post-chemotherapy miR-200c-3p tumor expression.

**Figure 4 cancers-14-02390-f004:**
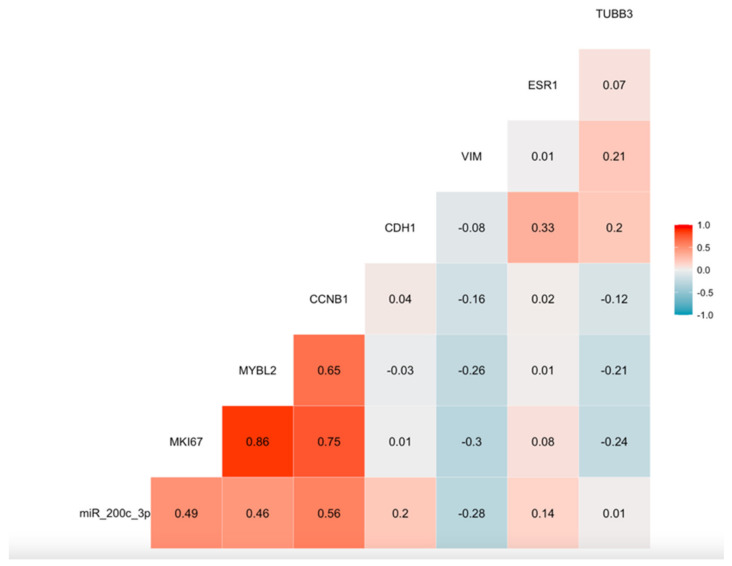
Correlation graph of miR-200c-3p and selected gene (proliferation, EMT, treatment resistance) expression in tumor core biopsies before chemotherapy. Spearman’s Rho coefficients are shown inside the graphic for each paired correlation.

**Table 1 cancers-14-02390-t001:** Clinical characteristics of all locally advanced breast cancer patients included in this study (*n* = 171).

Characteristics	Category	*n* (%)
Age, median, range		56 (21–79)
Histological type	Ductal	164 (95.9)
	Lobular	5 (2.9)
	Other	2 (1.2)
Histological grade	GI	8 (4.7)
	GII	64 (37.4)
	GIII	84 (49.1)
	N/A	15 (8.8)
Tumor phenotype	HR+ HER2−	78 (45.6)
	HR+ HER2+	34 (19.9)
	HR− HER2+	16 (9.3)
	Triple negative	39 (22.8)
	N/A	4 (2.3)
cT	cT1-2	84 (49.2)
	cT3	72 (42.1)
	cT4	14 (8.2)
	cTx	1 (0.6)
cN	cN0	54 (31.6)
	cN1	59 (34.5)
	cN2	34 (19.9)
	cN3	22 (12.9)
	cNx	2 (1.2)
Stage	IIA	35 (20.5)
	IIB	48 (28.1)
	IIIA	57 (33.3)
	IIIB	9 (5.3)
	IIIC	22 (12.9)
Pathologic complete response (pCR)	pCR (ypT0/is ypN0)	32 (18.9)
	pCR breast (ypT0/ypTis)	36 (21.1)

HR: hormone receptor; pCR: pathologic complete response.

**Table 2 cancers-14-02390-t002:** Clinical characteristics of metastatic breast cancer patients (*n* = 42).

Characteristics	Category	*n* (%)
Age, median (range)	59 (31–78)	
Histological Type	Ductal	35 (83,3)
	Lobular	7 (16,7)
Stage at diagnosis	M0 (recurrence)	9 (21,4)
	M1 (de novo metastatic disease)	33 (78,6)
Estrogen Receptor	Negative	6 (13,4)
	Positive	36 (85,7)
HER2-neu	Negative	33 (78,6)
	Positive	9 (21,4)
Immunohistochemical subtype	HR+ HER2−	31 (73,8)
	HR+ HER2+	6 (14,3)
	HR− HER2+	3 (7,1)
	TNBC	2 (4,8)
ECOG	0	13 (30.9)
	1	22 (52.4)
	2	4 (9.5)
	3	2 (4,7)
Metastatic site	Bone+/-Soft Tissues	23 (54.8)
	Visceral	7 (16.7)
	Visceral+Bone+/-Soff Tissues	12 (28.5)
	CNS	1 (2.4)
	N/A	1 (2.4)
Visceral Metastases	No	23 (54.8)
	Yes	19 (45.2)
Bone Metastases	No	7 (16.7)
	Yes	35 (83.3)
Treatment	Endocrine-based therapy	10 (23.8)
	Chemotherapy	18 (42.9)
	Chemotherapy + biological	13 (30.9)
	No treatment	1 (2.4)

ECOG: Eastern Cooperative Oncology Group performance status; HR: hormone receptor; pCR: pathologic complete response; TNBC: triple negative breast cancer.

**Table 3 cancers-14-02390-t003:** Univariate and multivariate Cox regression models for disease-free survival (DFS), distant relapse-free survival (DRFS) and overall survival (OS) of LABC patients (cohort A, *n* = 109) according to miR-200c-3p expression in tumor pre- and post-treatment.

	DFS		DRFS		OS	
UNIVARIATE ANALYSIS	HR (95% CI)	*p*-Value *	HR (95% CI)	*p*-Value *	HR (95% CI)	*p*-Value *
Grade 3	1.58 (0.74–3.38)	0.237	1.07 (0.52–2.19)	0.849	1.98 (0.83–4.74)	0.125
Stage III (vs. stage II)	1.59 (0.77–3.27)	0.211	2.42 (1.13–5.21)	0.024	1.82 (0.81–4.08)	0.147
cN2-3	2.04 (2.01–4.13)	0.047	3.12 (1.55–6.27)	0.001	2.54 (1.18–5.51)	0.018
Estrogen receptor positivity	0.69 (0.34–1.41)	0.311	1.27 (0.59–2.76)	0.537	0.70 (0.32–1.53)	0.376
HER2 amplification	0.99 (0.44–2.22)	0.988	1.02 (0.46–2.27)	0.965	0.91 (0.36–2.28)	0.842
Pathologic complete response (pCR)	0.27 (0.06–1.15)	0.076	0.14 (0.02–1.00)	0.050	0.16 (0.08–1.52)	0.163
miR-200c-3p (high vs. low), tumor, pre-NCT	1.82 (0.83–3.99)	0.134	2.27 (0.92–5.60)	0.074	2.76 (1.07–7.11)	0.027
miR-200c-3p (high vs. low), tumor, post-NCT	1.914(0.87–4.19)	0.105	2.77 (1.14–6.69)	0.024	3.15 (1.25–7.97)	0.015
MULTIVARIATE MODEL (with pre-NCT miR-200c-3p tumor expression)						
pCR	0.21 (0.05–0.92)	0.039	0.12 (0.01–0.89)	0.038	0.06 (0.01–0.49)	0.009
ER positivity	-----	-----	-----	-----	0.54 (0.21–1.40)	0.206
Grade 3	-----	-----	-----	-----	2.35 (0.85–6.54)	0.101
cN2-3	2.56 (1.19–5.52)	0.017	3.43 (1.46–8.00)	0.005	3.01 (1.21–7.52)	0.018
miR-200c-3p (high vs. low) pre-NCT	1.73 (0.79–3.79)	0.170	2.15 (0.87–5.29)	0.097	3.98 (1.49–10.60)	0.006
MULTIVARIATE MODEL (with post-NCT miR-200c-3p tumor expression)						
pCR	0.30 (0.07–1.33)	0.113	0.18 (0.02–1.39)	0.101	0.46 (0.10–2.06)	0.310
cN2-3	2.50 (1.16–5.41)	0.020	2.68 (1.18–6.07)	0.018	2.44 (1.08–5.50)	0.032
miR-200c-3p (high vs. low) post-NCT	1.52 (0.67–3.41)	0.313	2.06 (0.83–5.11)	0.118	2.63 (1.01–6.86)	0.049

* Cox regression; cut-off for classification of expression as high or low was the median value. DFS: disease-free survival; DRFS: distant relapse-free survival: ER: estrogen receptor; NCT: Neoadjuvant chemotherapy; OS: overall survival; pCR: pathologic complete response.

**Table 4 cancers-14-02390-t004:** Association of pre-treatment miR-200c-3p tumor expression with proliferation, EMT and treatment-resistance-related genes in LABC patients (cohort A).

	Pre-Chemotherapy				Post-Chemotherapy
	Total LABC	HR+ HER2−	HER2+	TNBC	Total LABC
	*n* = 105	*n* = 52	*n* = 28	*n* = 25	*n* = 77
MKI67					
Rho	0.49	0.44	0.41	0.66	0.38
*p*	<0.001	0.0067	0.055	0.004	0.003
MYBL2					
Rho	0.46	0.47	0.32	0.64	0.38
*p*	<0.001	0.003	0.142	0.006	0.003
CCNB1					
Rho	0.56	0.62	0.32	0.71	0.23
*p*	<0.001	<0.001	0.139	0.001	0.075
CDH1					
Rho	0.20	0.41	0.10	0.21	0.32
*p*	0.053	0.007	0.635	0.341	0.006
VIM					
Rho	−0.28	−0.42	−0.16	−0.08	−0.35
*p*	0.006	0.004	0.465	0.713	0.002
ESR1					
Rho	0.14	0.31	−0.01	0.10	−0.03
*p*	0.185	0.008	0.953	0.689	0.792
TUBB3					
Rho	0.01	0.16	−0.44	0.19	0.03
*p*	0.937	0.370	0.047	0.446	0.884

## Data Availability

The data that support the findings of this study are available from the corresponding author upon reasonable request.

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
