# Peer review of "Prognostic and Predictive Effects of Tumor and Plasma miR-200c-3p in Locally Advanced and Metastatic Breast Cancer"

_cancers, 2022, doi:10.3390/cancers14102390_

Round 1

Reviewer 1 Report

I have a few minor comments on the manuscript.

  1. 1, Line 20,: Please correct the sentence: “…..prognostic role of miR-200c-3p expression, both in plasma and in tumor” - the prognostic role of miR itself, since its expression in plasma does not occur.
  2. Is the change in expression of only two genes (VIM and CDH1) sufficient to their transcriptional co-response of EMT genes with miR-200c-3p expression? (Lines 21-23, P.21). I recommend to include Supplementary Figure S5. into the main text.
  3. The title of the Supplementary Figure S4. “Survival analysis of miR-200c-3p expression in pre-NCT and post-NCT plasma samples of LABC patients” does not correspond to the meaning of the graphs.
  4. If desired, the authors can add to the abstract their result that miR-200c-3p plasma levels were age-independent. This information may be of interest to a broader readership.

Author Response

  1. Page 1, Line 20,: Please correct the sentence: “…..prognostic role of miR-200c-3p expression, both in plasma and in tumor” - the prognostic role of miR itself, since its expression in plasma does not occur.
    • We have corrected this sentence and also line 28, eliminating the word "expression"
  2. Is the change in expression of only two genes (VIM and CDH1) sufficient to their transcriptional co-response of EMT genes with miR-200c-3p expression? (Lines 21-23, P.21). I recommend to include Supplementary Figure S5. into the main text.
    • We can not be sure of the relevance of the change in these two genes and more extensive expression data should be obtained to completely prove the impact on EMT programs. According to this comment, we have modified the sentences of lines 114-117 in page 15 of the Discussion section. 
    • Following Reviewer's suggestion, we have included the Figure S5 as a main figure (Figure 4) in the revised manuscript.
  3. The title of the Supplementary Figure S4. “Survival analysis of miR-200c-3p expression in pre-NCT and post-NCT plasma samples of LABC patients” does not correspond to the meaning of the graphs.
    • We apologize for this mistake. The correct title has been added to Supplementary Figure S4
  4. If desired, the authors can add to the abstract their result that miR-200c-3p plasma levels were age-independent. This information may be of interest to a broader readership.
    • Following Reviewer's suggestion, we have added this information to the abstract.

Reviewer 2 Report

The authors have made significant changes in the manuscript, including toning down the interpretation of the data and adding new information/data. Overall, they have adequately addressed all the (many) issues that were raised.

Author Response

We would like to thank the Reviewer for appreciating the changes made in the original manuscript and for his contribution to improving it. 

Reviewer 3 Report

The authors replied to the questions arised

Author Response

We are grateful to the Reviewer for his contribution to the improvement of the original manuscript.